# Highly Oxygenated Organic Molecules Produced by the Oxidation of Benzene and Toluene in a Wide Range of OH Exposure and NOx Conditions

Xi Cheng,[1] Qi Chen,[1,*] Yong Jie Li,[2,*] Yan Zheng,[1] Keren Liao,[1] Guancong Huang[1]

[1]State Key Joint Laboratory of Environmental Simulation and Pollution Control, BIC-ESAT and IJRC, College of Environmental Sciences and Engineering, Peking University, Beijing, China

[2]Department of Civil and Environmental Engineering, Faculty of Science and Technology, University of Macau, Taipa, Macau, China

*Correspondence to*: Qi Chen (qichenpku@pku.edu.cn) and Yong Jie Li (yongjieli@um.edu.mo)

**Abstract.** Oxidation of aromatic volatile organic compounds (VOCs) leads to the formation of tropospheric ozone and secondary organic aerosol, for which gaseous oxygenated products are important intermediates. We show herein experimental results of highly oxygenated organic molecules (HOMs) produced by the oxidation of benzene and toluene in a wide range of OH exposure and $NO_x$ conditions. The results suggest multi-generation OH oxidation plays an important role in the product distribution, which likely proceeds more preferably via H subtraction than OH addition for early-generation products from light aromatics. More oxygenated products present in our study than in previous flow-tube studies, highlighting the impact of experimental conditions on product distributions. The formation of dimeric products however was suppressed and might be unfavorable under conditions of high OH exposure and low $NO_x$ in toluene oxidation. Under high-$NO_x$ conditions, nitrogen-containing multifunctional products are formed, while the formation of other HOMs is suppressed. Products containing two nitrogen atoms become more important as the $NO_x$ level increases, and the concentrations of these compounds depend significantly on $NO_2$. The highly oxygenated nitrogen-containing products might be peroxyacylnitrates, implying a prolonged effective lifetime of $RO_2$ that facilitates regional pollution. Our results call for further investigation on the roles of high-$NO_2$ conditions in the oxidation of aromatic VOCs.

# 1 Introduction

Atmospheric oxidation of volatile organic compounds (VOCs) are crucial in the formation of tropospheric ozone ($O_3$) and secondary organic aerosol (SOA) (Calvert et al., 2015; Seinfeld and Pandis, 2016). Enormous studies have been conducted for the $O_3$ formation potentials and SOA yields of light aromatic VOCs such as benzene and toluene (Calvert et al., 2002; Atkinson and Arey, 2003; Ziemann and Atkinson, 2012). The peroxy radicals (either $RO_2$ or $HO_2$) that are generated from the oxidation process convert NO to $NO_2$. Ground-state oxygen atoms $O(^3P)$ are produced through the photolysis of $NO_2$, and the reaction

between the $O(^3P)$ and $O_2$ is the main source of tropospheric $O_3$ (Calvert et al., 2015). The oxygenated organic products, on the other hand, may form SOA through either nucleation or condensation with various mass yields, depending on the structure of the precursors and the $NO_x$ (= NO + $NO_2$) level (Ng et al., 2007; Li et al., 2016). Therefore, for both $O_3$ and SOA formation via VOC oxidation, $NO_x$ plays critical roles (Atkinson, 2000; Sato et al., 2012; Tsiligiannis et al., 2019; Garmash et al., 2020).

OH-initiated oxidation of light aromatics occurs mainly via OH addition, with about 90% of preference (Calvert et al., 2002;

Wu et al., 2014; Schwantes et al., 2017). As described in Sect. S1 in the Supplement, the formation of bicyclic peroxy radicals (BPRs) is central to aromatic oxidation in the absence of $NO_x$. Significant fractions of the oxidation (e.g., 0.35 for benzene and 0.65 for toluene) may lead to the formation of BPRs (Scheme S1 in the Supplement) (Birdsall et al., 2010), followed by further reactions to form highly oxygenated organic molecules (HOMs) (Crounse et al., 2013; Ehn et al., 2014; Berndt et al., 2016; Bianchi et al., 2019). The fate of the BPRs has been recently investigated by using time-of-flight chemical ionization

mass spectrometer (TOF-CIMS), which is suitable for measuring HOMs. The TOF-CIMS measurements show that major products for the reactions between BPRs and $HO_2$/$RO_2$ in the absence of $NO_x$ include carbonyls, alcohols, hydroperoxides, dimers, and alkoxy (RO) radicals (Birdsall et al., 2010; Wang et al., 2017; Molteni et al., 2018; Zaytsev et al., 2019b; Garmash et al., 2020). The BPR-derived products that still possess the bicyclic skeleton are considered as the major ring-retaining products from the oxidation of light aromatics (Zaytsev et al., 2019b). Decomposition of RO radicals may lead to fragmented

products such as di-carbonyls and epoxides (Yu and Jeffries, 1997; Yu et al., 1997; Arey et al., 2009; Zaytsev et al., 2019b). The formation of HOMs may involve multi-step auto-oxidation and multi-generation OH reaction (Zaytsev et al., 2019b; Garmash et al., 2020; Y. Wang et al., 2020). Yet, it is still unclear whether the formation of HOMs is controlled by multi-step auto-oxidation or multi-generation OH oxidation.

Conditions of flow-tube or smog-chamber experiments have covered various conditions of OH concentrations ($10^4$ to $10^{11}$

molecule $cm^{-3}$) and residence times (10 s to 1 h) (Wang et al., 2017; Molteni et al., 2018; Garmash et al., 2020). Extrapolation of these results to tropospheric conditions requires further investigations under a wide range of $NO_x$ levels. $NO_x$ are rich in urban environments and compete with $HO_2$ and $RO_2$ for the termination of $RO_2$ radicals (Calvert et al., 2002; Seinfeld and Pandis, 2016). Early studies show a strong dependence of SOA mass yields on $NO_x$, owing to the formation of more-volatile products through the termination of $RO_2$ by NO (Ng et al., 2007; Sato et al., 2012). There are also a few recent studies on the

gaseous oxygenated products from aromatic oxidation in the presence of $NO_x$ (Tsiligiannis et al., 2019; Garmash et al., 2020; Y. Wang et al., 2020). Garmash et al. (2020) found that nitrated phenols (NPs) contribute significantly to the gaseous nitrogen-

containing HOMs produced by the benzene oxidation in the presence of $NO_x$. Tsiligiannis et al. (2019) show prevalent formation of organic nitrates (ONs) from 1,3,5-trimethylbenzene (TMB) oxidation, especially for atmospherically relevant $[NO_x]:[VOC]$ ratios of greater than 1. In addition to ONs, Y. Wang et al. (2020) show the elevated formation of dinitrates from the oxidation of TMB isomers in the presence of $NO_x$. In general, the formation of these nitrogen-containing products suppresses the formation of other HOMs (Tsiligiannis et al., 2019; Garmash et al., 2020; Y. Wang et al., 2020). However, the dependence of product distributions on $NO_x$ conditions (e.g., $[NO_x]:[VOC]$ or $[NO_2]:[NO]$ ratios) remains largely unclear.

In this study, we investigate the production of gaseous HOMs from the OH-initiated oxidation of benzene and toluene in an oxidation flow reactor (OFR) by using nitrate-adduct TOF-CIMS ($NO_3^-$-TOF-CIMS). A wide range of OH exposure and $NO_x$ conditions are studied. Distributions and molar yields of key products are investigated. Kinetic analysis helps inferring the possible formulas of nitrogen-containing HOMs.

## 2 Experimental Section

### 2.1 Oxidation flow reactor

The oxidation experiments are conducted in a 13.3-L Aerodyne Potential Aerosol Mass (PAM) OFR reactor. The schematic of the experimental setup and the example experimental sequence are shown in Figs. S1 and S2 in the Supplement, respectively. The OFR was operated in two continuous-flow modes named as OFR254-5 and OFR254-5-i$N_2$O (Lambe et al., 2017; Peng et al., 2018). For the OFR254-5 mode, the inside UV lamps emit photons at 254 nm to generate OH radicals in the reactor via the reaction of $O(^1D) + H_2O \rightarrow 2OH$. An outside UV lamp produces $O_3$ and leads to about 5 ppm of $O_3$ in the OFR. For the OFR254-5-i$N_2$O mode, $N_2O$ (99.5%) is injected into the OFR to achieve $N_2O$ mixing ratios of 1.1% (OFR254-5-i$N_2$O1.1) and 4.4% (OFR254-5-i$N_2$O4.4). NO and $NO_2$ are then formed by the reaction $O(^1D) + N_2O \rightarrow 2NO$, followed by the reaction $NO + O_3 \rightarrow NO_2 + O_2$ (Lambe et al., 2017). The conditions for OFR254-5 are considered as low-$NO_x$, whereas the latter are high-$NO_x$. Under each of these three sets of experiments, OH exposure and $NO_x$ levels were varied by ramping the voltage of the UV lamps in the OFR. The total flow rate was about 8.4 L min$^{-1}$, resulting in a mean residence time of 95 s in the OFR. Relative humidity (RH) in the OFR was about 20 to 55% at $25 \pm 2$ °C, corresponding to $H_2O$ mixing ratios of about 0.7 to 1.5%. Details about the experimental setup are provided in Sect. S2 in the Supplement.

In total, 28 experiments were conducted for initial mixing ratios of 110 ppb benzene and 50 ppb toluene. Table S1 in the Supplement lists the experimental conditions as well as measured and derived quantities for all experiments. The concentrations of reactive species such as OH, $HO_2$, NO, and $NO_2$ were estimated by an OFR-based photochemical box model (PAMchem) (Lambe et al., 2017). The box model simulations are described in detail in Sect. S2, for which calibration experiments with $SO_2$ was conducted to constrain the actinic flux at 254 nm (Fig. S3 in the Supplement). Briefly, the model suggests OH exposure of about $1.1 \times 10^{11}$ to $2.5 \times 10^{12}$ molecules cm$^{-3}$ s for our experiments, corresponding to equivalent photochemical age of 0.8 to 19.3 days with a mean OH concentration of $1.5 \times 10^6$ molecules cm$^{-3}$. For high $NO_x$ experiments,

the OFR-exit NO and $NO_2$ concentrations were 0.2 to 5.1 ppb and 15.8 to 231.4 ppb, respectively, leading to [$NO_x$]:[VOC] ratios of 0.2 to 2.2 for benzene and 0.4 to 4.7 for toluene. The OFR-based photochemical box simulations show that the aromatic oxidation reactions in our experimental conditions were dominated by OH rather than $O_3$ (Lambe et al., 2011; Peng et al., 2016). Similar to other OFR studies, we think the reaction rates of $O_3$ with oxidation products that contain double bounds are likely slower compared with that of OH (Molteni et al., 2018; Y. Wang et al., 2020). The $NO_3$ concentrations in the OFR range from 0.01-0.09 ppb. The general reaction rate of $RO_2$ with OH is one order of magnitude greater than that with $NO_3$ under our experimental conditions (Jenkin et al., 2019). For such rates, the $NO_3$ reactions with HOM products might be minor. The $NO_3$ oxidation of phenols however may contribute efficiently to the formation of nitrated phenols in the OFR because of the high branching ratio.

## 2.2 Chemical characterization

HOMs were characterized by an Aerodyne $NO_3^-$-TOF-CIMS. Details about the instrument operation and data analysis are described elsewhere (Cheng et al., 2021). Briefly, mass calibration was performed on the reagent ions and selected Teflon-related ions, which covers the $m/z$ range of 62 to 676 with mass accuracies of less than 10 ppm. The "non-production" ions were identified by positive matrix factorization (PMF) analysis and were removed from the analysis (Fig. S4 in the Supplement). The $HNO_3NO_3^-$-adduct ions were identified by the signal correlations between the $NO_3^-$-adduct and $HNO_3NO_3^-$-adduct ions and were removed from the analysis (Fig. S5 in the Supplement). Only $NO_3^-$ adduct ions are presented herein. The background signals of individual ions were determined on the basis of the measurements made without the injection of VOCs (Fig. S2). Because of the lack of standards, we applied the calibration factor of 4-nitrophenol (i.e., 0.0020 ncps ppt$^{-1}$ or $1.66 \times 10^{10}$ molecules cm$^{-3}$) to HOMs, which is similar to the commonly-used calibration factors of $H_2SO_4$ (i.e., $1.62 \times 10^{10}$ molecules cm$^{-3}$ for our instrument herein and $1.89 \times 10^{10}$ molecules cm$^{-3}$ reported by Jokinen et al. (2012)) and perfluoroheptanoic acid (i.e., $1.6 \times 10^{10}$ molecules cm$^{-3}$ reported by Ehn et al. (2014)). All calibration factors are corrected by wall losses. Ehn et al. (2014) reported a $\pm 50\%$ of uncertainty. For our experiments, we estimate an uncertainty of 42% on the basis of nitrated phenol calibrations (Cheng et al., 2021). In some experiments, the VOC precursors and less-oxygenated gaseous products were monitored by an IONICON proton transfer reaction-quadrupole interface time-of-flight mass spectrometer (PTR-QiTOF). The instrument operation and data analysis have been described previously (Huang et al., 2019). The measurements have a total uncertainty of about 21%. Particle size distributions were measured by a scanning mobility particle sizer (SMPS, TSI, 3938). The SOA mass concentrations are measured by an Aerodyne long-time-of-flight soot-particle aerosol mass spectrometer (LTOF-SP-AMS) for the calculation of condensation sink in wall-loss corrections of HOMs (Zheng et al., 2020).

## 3 Results and discussion

### 3.1 Product categories

Figure 1 shows the mass spectra of gaseous oxygenated products produced by benzene and toluene oxidation in typical low- and high-$NO_x$ experiments. The fitted ion peaks are categorized into fragmented products, open-shell monomeric products, closed-shell monomeric products, and dimeric products as well as nitrogen-containing products when $NO_x$ is present. The proposed products categories are based on traditional understanding and recent mechanistic developments of aromatic oxidation (Wang et al., 2017; Mentel et al., 2015; Molteni et al., 2019). In benzene and toluene ($C_xH_y$) oxidation by OH radicals, addition of OH and two $O_2$ molecules lead to the formation of BPR $C_xH_{y+1}O_5$, which increase the molecular composition by one H atom and five oxygen atoms. The BPR $C_xH_{y+1}O_5$ can undergo further autoxidation and form radicals with formula $C_xH_{y+1}O_7$. Autoxidation of $RO_2$ radicals involves intramolecular hydrogen shifts to the peroxide group from other carbon atoms and subsequent addition of oxygen to the produced carbon-centered radicals. The H shift itself does not modify the molecular composition, but $O_2$ addition increases the oxygen content by an even number. Besides, BPR $C_xH_{y+1}O_5$ can form termination products of carbonyls ($C_xH_yO_4$), alcohols ($C_xH_{y+2}O_4$), hydroperoxides ($C_xH_{y+2}O_5$), dimers and nitrogen-containing compounds. Notably, products distributions are affected by the numbers of generation of auto-oxidation pathway and the H-shift reaction rates. Some product formulae may involve multiple pathways as discussed later in Sect. 3.2 for which additional information (e.g., collision induced dissociation experiments) may be helpful to further explore possible structures of the HOM molecules (Zaytsev et al., 2019a).

Tables S2-S3 in the Supplement list the corresponding peak lists and relative signal contributions of major products in each category for the low- and high-$NO_x$ experiments in Fig. 1. The fragmented products are the most abundant category under low-$NO_x$ conditions. They have carbon numbers less than their precursors, indicating possibly ring opening/scission from presumably RO radicals (Zaytsev et al., 2019b). In addition, fragmented products can be formed through CO elimination from an acyl radical (Rissanen et al., 2014), splitting $CO_2$ from an RO radical (Garmash et al., 2020), or dealkylation (Birdsall and Elrod, 2011). In this study, $C_2H_4O_4$ shows the highest signal intensity in the fragmented category under low-$NO_x$ conditions. $C_4H_4O_5$ and $C_5H_4O_6$ are the other two abundant common products in this category. Under high-$NO_x$ conditions, $C_4H_2O_5$ is the most abundant common product for benzene and toluene oxidation. Many of the fragmented products have been detected in SOA (Gallimore et al., 2011; Gowda and Kawamura, 2018).

The ring-retaining HOM monomers may have even (closed-shell) or odd (open-shell) hydrogen numbers (Molteni et al., 2018; Zaytsev et al., 2019b; Garmash et al., 2020). With relatively large carbon numbers, the $RO_2$ radicals generally have lifetimes of seconds that are much longer than the RO radicals of $< 10^{-4}$ s (Orlando et al., 2003; Seinfeld and Pandis, 2016; Zhao et al., 2018). The open-shell monomeric products observed by the $NO_3^-$-TOF-CIMS in this study are therefore more likely $RO_2$ radicals rather than RO radicals. Under low-$NO_x$ conditions, the BPRs have relatively low signal intensities (0.1%). Products that are presumably formed by further auto-oxidation of BPRs such as $C_6H_7O_7$ and $C_6H_7O_9$ for benzene (or $C_7H_9O_7$ and $C_7H_9O_9$

for toluene) have much greater signal intensities compared with the BPRs, especially for the $O_9$ products. Other open-shell monomeric products with two less hydrogen atoms (e.g., $H_5$ for benzene and $H_7$ for toluene) or with an even number of oxygen atoms (e.g., $O_6$ or $O_{10}$) are also present. Under high-$NO_x$ conditions, the main open-shell monomers are the $O_9$ products, similar to the low-$NO_x$ case.

Among the closed-shell monomeric products produced by benzene oxidation, $C_6H_6O_{5-10}$ and $C_6H_8O_{5-10}$ with one oxygen atom apart show relatively high signal intensities, and $C_6H_6O_5$ and $C_6H_6O_6$ are the most abundant ones. $C_6H_6O_5$ and $C_6H_6O_6$ might be carbonyl products from the termination of $C_6H_7O_6$ and $C_6H_7O_7$ by $HO_2$ or $RO_2$ (Zaytsev et al., 2019b; Garmash et al., 2020). $C_6H_7O_6$ is probably formed by the ring breakage of bicyclic alkoxy radical followed by the 1,5 aldehydic alkoxy H-shift reactions (Xu et al., 2020), which involves the RO pathway as described in detail in Sect. S3 of the Supplement. For other products formed by the termination reactions of BPR with $HO_2$ or $RO_2$, bicyclic hydroperoxide ($C_6H_8O_5$), carbonyl ($C_6H_6O_4$) and alcohol ($C_6H_8O_4$) have relatively small signals, whereas the products that may involve two or three steps of auto-oxidation (e.g., $C_6H_8O_7$ and $C_6H_8O_9$) have greater signal intensities. BPRs (i.e., $C_6H_7O_5$ from benzene oxidation and $C_7H_9O_5$ from toluene oxidation) can undergo unimolecular isomerization reactions to form more oxidized peroxy radicals, which compete with bimolecular reactions (Wang et al., 2017). The relative greater signal intensities of ring-retaining $O_7$ or $O_9$ HOM monomers compared with $O_{<6}$ ones suggest that the termination of $RO_2$ by $HO_2$ and the phenol oxidation pathway are perhaps relatively less important than the auto-oxidation pathway under experimental conditions herein (Calvert et al., 2002; Schwantes et al., 2017; Garmash et al., 2020). Similarly for toluene oxidation, the bicyclic hydroperoxide ($C_7H_{10}O_5$) shows lower signals than $C_7H_{10}O_7$ and $C_7H_{10}O_9$ that may be associated with multiple steps of auto-oxidation. $C_7H_8O_6$ and $C_7H_8O_7$ are the main closed-shell monomers that might be carbonyls that are produced by the termination of $C_7H_9O_7$ and $C_7H_9O_8$ by $HO_2$ or $RO_2$, respectively. The formation of $C_7H_8O_7$ may also be explained by the RO pathway, similar to $C_6H_6O_5$ (Sect. S3). The RO pathway, if it occurs, may potentially lead to various products with significant signals, which remains largely overlooked in studies of aromatic oxidation (Xu et al., 2020).

The other two categories are dimeric and nitrogen-containing products. In our study, the ion intensities of dimeric products with odd hydrogen atoms are low. We therefore focus on the dimeric products with even hydrogen atoms. Under low-$NO_x$ conditions, dimeric products with 8-14 even oxygen atoms are clearly present. $C_{12}H_{14}O_8$ and $C_{14}H_{18}O_8$ are perhaps formed via self-reactions of the BPRs for benzene and toluene, respectively. The distribution of dimeric products is in line with the monomeric open-shell and closed-shell products. We observe a number of dimeric products with odd oxygen numbers that are perhaps produced by cross-reactions of odd- and even-oxygen $RO_2$ (Molteni et al., 2018; Garmash et al., 2020). The presence of $NO_x$ results in the formation of nitrogen-containing products with one or two nitrogen atoms, and the signal abundances of all other oxygenated products are much lower than the case of low-$NO_x$ experiments. Such a suppression has been reported previously (Tsiligiannis et al., 2019; Garmash et al., 2020; Y. Wang et al., 2020; Mehra et al., 2020). The nitrogen-containing products are expected to be organic nitrates or nitrated phenols. For benzene oxidation, $C_6H_5NO_3$, $C_6H_5NO_4$, and $C_6H_4N_2O_6$ are the most abundant, which are plausibly nitrophenol, nitrocatechol, and dinitrocatechol, respectively. For toluene, $C_6H_5NO_3$

(nitrophenol), $C_6H_5NO_4$ (nitrocatechol), $C_7H_7NO_3$ (methylnitrophenol), and $C_7H_7NO_4$ (methylnitrocatechol) are the main tentatively assigned nitrated phenols. Significant secondary production of these compounds from aromatic oxidation have been observed in urban Beijing (Cheng et al., 2021). Other products such as $C_6H_7NO_8$, $C_6H_7NO_9$, and $C_7H_9NO_8$ are most likely organic (peroxy) nitrates. Schemes S2 and S3 in the Supplement give examples of the proposed formation mechanisms for nitrogen-containing products and multi-step auto-oxidation products starting from the BPR of $C_7H_9O_5$ (Sect. S1).

### 3.2 Low-NO$_x$ conditions

*Product Distribution.* The products that we observed herein (Tables S2-S3) are generally in agreement with those found in previous low-NO$_x$ studies (Schwantes et al., 2017; Molteni et al., 2018; Zaytsev et al., 2019b; Mehra et al., 2020; Garmash et al., 2020). Differences exist in the relative abundance of species with different oxygen contents within each product category (Molteni et al., 2018; Garmash et al., 2020). Table S4 in the Supplement lists the experimental conditions and relative signal intensities of some major oxygenated products formed by benzene oxidation. Molteni et al. (2018) observes predominant production of $C_6H_8O_5$ (plausibly hydroperoxides) and $C_{12}H_{14}O_8$. Garmash et al. (2020) shows relatively high signals of $C_6H_8O_9$ and $C_{12}H_{14}O_8$ in the flow tube experiments, whereas the chamber study and our study both show relatively high signals of $C_6H_8O_7$ and $C_6H_8O_9$ and lower signals of dimeric products. Moreover, the signal intensities of RO$_2$ (e.g., $C_6H_7O_5$ and $C_6H_7O_7$) are greater in our study than in other studies. Both experimental conditions and instrument detection are factors that may affect the product distribution. For example, a longer residence time might promote multi-step auto-oxidation. For auto-oxidation, H shift has rate constants of about $10^{-3}$ to 1 s$^{-1}$ and is likely the rate-determining step under atmospheric conditions, having reaction times of 1 to $10^3$ s that are much longer than the subsequent $O_2$ addition of μs (Orlando et al., 2003; Bianchi et al., 2019). Garmash et al. (2020) indicated that fewer auto-oxidation steps would be expected in flow-tube experiments. The flow-tube study herein has relatively longer residence time than others, which is consistent with relatively more abundant $O_7$ and $O_9$ products. Ambient environments have much lower OH concentrations but longer residence times depending on meteorological conditions, which may suggest more oxygenated products from multi-steps of auto-oxidation. Alternatively, the differences in HO$_2$ and RO$_2$ concentrations among different studies that remain unclear might affect the extent of auto-oxidation. For detection, the instrument efficiency might be different for HOMs having different clustering capability (e.g., numbers of functional group as hydrogen-bond donors) (Hyttinen et al., 2015). Tuning might affect the transmission of product ions in different *m/z* range or affect the efficiency of dimer clustering (Heinritzi et al., 2016; Brophy and Farmer, 2015).

*Effects of OH Exposure.* As shown in Fig. 2a, the concentrations of fragmented, monomeric closed-shell and open-shell, dimeric products formed by benzene oxidation increase with increasing OH exposure that corresponds to 2.4 to 16.1 days of atmospheric equivalent photochemical age. Elevated concentrations of monomeric open-shell products (e.g., RO$_2$) and HO$_2$ are expected for greater OH exposure, leading to enhanced production of the monomeric closed-shell and dimeric products through the RO$_2$ + HO$_2$ and RO$_2$ + RO$_2$ reactions. These processes may be limited by the availability of RO$_2$ (< 0.9 ppt in our study) rather than that of HO$_2$ (0.5 - 2.4 ppb) (Table S1). Consistently, the concentrations of dimeric products are much lower

than that of monomeric closed-shell products. For toluene oxidation, the dependence of the product formation on OH exposure is less significant than that for benzene (Fig. 2b). The concentrations of fragmented, monomeric closed-shell and open-shell products first increase and then slightly decrease with the increasing OH exposure that corresponds to 2.4 to 19.4 days of atmospheric equivalent photochemical age. Interestingly, unlike benzene oxidation, the concentrations of dimeric products decrease as the OH exposure increases, suggesting perhaps unfavorable dimer formation under conditions of high OH exposure for toluene oxidation. One potential contributor to the difference of the dependence of dimer formation on OH exposure is the more significant elevated $RO_2$ concentrations (0.2 to 0.9 ppt) as the OH exposure increases for benzene oxidation than that (0.3 to 0.5 ppt) for toluene oxidation while the $HO_2$ concentrations in the two sets of experiments are similar (1.5-2.4 ppb). The enhancement of $RO_2$ concentrations may promote the dimer formation through the self or cross reactions of $RO_2$ (Mohr et al., 2017). On the other hand, a previous study indicates that the accretion of $RO_2$ depends on the functional groups of the $RO_2$ (Berndt et al., 2018). Whether the decreasing concentrations of dimeric products with OH exposure is related to the steric effects of the substituted methyl group of toluene requires further investigations. The overall atomic oxygen-to-carbon (O:C) ratios of HOM products increase from 1.18 to 1.26 for benzene oxidation and from 1.16 to 1.23 for toluene oxidation as the OH exposure increases, suggesting more oxygenated product distributions as detected. The overall atomic hydrogen-to-carbon (H:C) ratios of HOMs are 1.10-1.14 for benzene oxidation and 1.14-1.18 for toluene oxidation, and the changes with OH exposure are negligible.

Figure 2c-d shows the concentrations of individual representative open-shell and closed-shell monomers for increasing OH exposures. The open-shell monomers are mainly $C_xH_{y+1}O_5$ (BPRs) and $C_xH_{y+1}O_7$ ($RO_2$ formed from one more step of auto-oxidation). The concentrations of $C_xH_{y+1}O_7$ are approximately one order of magnitude higher than the concentrations of $C_xH_{y+1}O_5$. As described in Sect. 3.1, the further reactions of $C_xH_{y+1}O_5$ and $C_xH_{y+1}O_7$ may form closed-shell monomers such as hydroperoxides ($C_xH_{y+2}O_{5,7}$), carbonyls ($C_xH_yO_{4,6}$), and alcohols ($C_xH_{y+2}O_{4,6}$) (Mentel et al., 2015; Molteni et al., 2019), although these formulae may correspond to other functionalities depending on the reaction rates and numbers of generation of autoxidation pathways. For multiple steps of auto-oxidation of BPRs or $RO_2$ radicals ($C_xH_{y+1}O_5$ or $C_xH_{y+1}O_7$), the products with two more hydrogen atoms than the precursor are plausibly hydroperoxides ($C_xH_{y+2}O_z$) if $z$ is an odd number and alcohols if an even number. On the other hand, products that have the same hydrogen atoms ($C_xH_yO_z$) are likely carbonyls with an even number of $z$. Instead, if the carbonyl formation involves the RO pathway, $C_xH_yO_z$ with an odd number of $z$ may be formed. The concentrations of $C_xH_{y+2}O_7$ and $C_xH_yO_6$ originated from $C_xH_{y+1}O_7$ are 1 to 2 orders of magnitudes greater than the concentrations of $C_xH_{y+2}O_5$ and $C_xH_yO_4$ originated from $C_xH_{y+1}O_5$. The greater signals of $O_7$ products suggest our experimental conditions perhaps favor the formation of more-oxygenated products.

Enhanced formation of more oxygenated products was observed for elevated OH exposures. For example, the concentrations of $C_6H_7O_7$ increase first and stay relatively stable as the OH exposure increases, whereas the concentrations of $C_6H_7O_5$ decreases at high OH exposures for benzene oxidation. The concentrations of the hydroperoxide products ($C_xH_{y+2}O_{5,7}$) such as $C_6H_8O_7$ increases as OH exposure increases whereas the concentrations of $C_6H_8O_5$ decreases. For toluene oxidation, the

concentrations of $C_xH_{y+2}O_{5,7}$ ($C_7H_{10}O_5$ and $C_7H_{10}O_7$) both decrease. The concentrations of carbonyl products ($C_xH_yO_{4,6}$) increase with OH exposure for benzene but not for toluene. We do not observe significant signals for $C_xH_{y+2}O_4$ (alcohols) from BPR $C_xH_{y+1}O_5$, but $C_xH_{y+2}O_6$ from $RO_2$ $C_xH_{y+1}O_7$ are of high concentrations. The concentrations of $C_6H_8O_6$ (alcohol) from benzene also increase as the OH exposure increases while the concentrations for $C_7H_{10}O_6$ from toluene decrease. Overall, the enhanced formation of more-oxygenated products at high OH exposure is more significant for benzene than for toluene. A possible explanation is that toluene oxidation may involve more multi-generation OH oxidation because of the substituted methyl group.

Increasing OH exposure also enhances the formation of more-oxygenated fragmented products (Fig. S6 in the Supplement). $C_4H_4O_3$ (perhaps epoxybutanedial) has been widely observed in aromatic oxidation (Wang et al., 2013; Wu et al., 2014; Zaytsev et al., 2019b). The concentrations of $C_4H_4O_3$ decrease monotonically as the OH exposure increases for both benzene and toluene oxidation. For toluene oxidation, $C_3H_4O_2$ (perhaps methylglyoxal), $C_4H_4O_2$ (perhaps butenedial) and $C_5H_6O_2$ (perhaps methylbutenedial) that were detected by the PTR-QiTOF also show similar monotonic decreases. By contrast, the concentrations of $C_3H_4O_5$, $C_4H_4O_5$, and $C_5H_4O_6$ increase with increasing OH exposures for both benzene and toluene oxidation. The concentrations of other $O_{5-7}$ products also increase as the OH exposure increases. Similar to the monomeric open-shell and closed-shell products, the increasing trends are more significant for benzene-derived products than those for toluene-derived ones. This observation is in agreement with Garmash et al. (2020) who suggested that first-generation fragmented products were converted to more-oxygenated ones through further oxidation as the OH exposure increases. Alternatively, these more-oxygenated fragmented products might also be direct decomposition products of highly oxygenated RO radicals, which have been shown to play important roles in product formation (Noda et al., 2009; Birdsall and Elrod, 2011).

*Multi-generation OH oxidation.* In benzene and toluene ($C_xH_y$) oxidation by OH radicals, we may expect that one OH addition could lead to termination products with hydrogen numbers of $y$ and $y+2$. When the second OH attack occurs, products with hydrogen numbers $< y$ and $> y+2$ may be produced by multi-generation OH reactions (Table S5 in the Supplement). For example, the closed-shell monomeric products from benzene oxidation should contain at least one double bond which can further react with OH radicals via OH addition to form products with two more hydrogen atoms (e.g., to form $C_6H_{10}O_z$). The formation of $C_6H_4O_z$ from the closed-shell monomeric products of benzene oxidation may involve multi-generation OH reactions via H abstraction and subsequent termination by a loss of OH or $HO_2$ (Molteni et al., 2018; Garmash et al., 2020). There are significant contributions of products with hydrogen numbers $< y$ and $> y+2$ in our experiments (Tables S2 and S3), highlighting the importance of multi-generation OH reactions in production formation. On the other hand, the formation of $y$ and $y+2$ products may also involve multi-step OH addition or H subtraction that make conversions between the two groups of products. For the phenolic pathway, the main products of $C_6H_6O_z$ and $C_7H_8O_z$ may be produced by dihydroxy-, trihydroxy- and even multi-OH substituted-benzene or toluene (Schwantes et al., 2017). The contributions of a third OH attack to the total of detected signals are often very low (Molteni et al., 2018).

Figure 3 shows the concentrations and the relative contributions of the closed-shell monomeric product groups including $C_xH_{y-2}O_z$, $C_xH_yO_z$, $C_xH_{y+2}O_z$ and $C_xH_{y+4}O_z$. For both benzene and toluene oxidation, the concentrations of these monomers increase as the OH exposure increases for the atmospherically relevant equivalent photochemical age (< 10 days). For greater OH exposures, the concentrations of $y$-2 products keep increasing, whereas the concentrations of other products start to decrease (Fig. 3a-b). The decreasing trends at high OH exposures are more significant for toluene-derived products. Such differences might be explained by (1) the faster OH + VOC rate for toluene than for benzene and (2) the methyl group in toluene as well as the added -OH groups during oxidation that may increase the electron density on the aromatic ring through a resonant electron-donating effect and thereby activate the aromatic ring and facilitate further addition reactions to form products with multiple -OH groups (M. Wang et al., 2020). Interestingly, the relative fractions of the $C_xH_{y-2}O_z$ and $C_xH_{y+4}O_z$ products for both benzene and toluene oxidation show opposite trends (Fig. 3c-d). The increasing fractions of $y$-2 products but decreasing fractions of $y$+4 products for increasing OH exposures as well as the monotonically increasing concentration of $y$-2 products suggest that the multi-generation OH oxidation may proceed preferably via H subtraction rather than OH addition according to current mechanistic understanding of aromatic oxidation. Garmash et al. (2020) also noted that higher OH concentrations caused a larger variety in HOM products, with H-abstraction oxidation becoming possibly more significant. Compared with their precursors (benzene and toluene), the closed-shell monomeric products are less conjugated and thus OH addition is probably less favorable.

***Estimated Molar Yields.*** Figure S7 in the Supplement shows the scatter plot of the concentrations of HOMs detected by $NO_3^-$-TOF-CIMS and the VOC oxidation rate for benzene and toluene oxidation under low-$NO_x$ conditions. As expected, the product concentrations increase with the VOC oxidation rates. To calculate the molar yields of HOM products, wall losses are corrected, including the loss in the sampling line, the loss to the OFR walls estimated by eddy diffusion, the loss to aerosol particles in the OFR (i.e., the condensation sink calculated on the basis of particle measurements), and the loss to non-condensable products due to continuous reaction with OH (Palm et al., 2016). Details are provided in Sect. S4 in the Supplement. Our estimated molar yields for the HOM products are 0.22 ± 0.10% (mean ± one standard deviation) for benzene oxidation and 0.46 ± 0.20% for toluene oxidation. These yields are much lower than the smog-chamber results of 4.1% to 14.0% for benzene oxidation reported by Garmash et al. (2020) but slightly greater than the flow-tube yields of 0.1% to 0.2% reported by Molteni et al. (2018). A key difference of the experimental conditions is the much longer residence time in the chamber study (Table S4), suggesting perhaps a long characteristic time of HOM formation from aromatic oxidation. The study of Molteni et al. (2018) only provided the initial OH concentration as listed in Table S4 that should decrease significantly as the reaction proceeded, for which we cannot rule out the possibility of low OH exposure that leads to fewer oxidation steps (i.e., lower yields of HOMs). Instrument sensitivity might also affect the detection of HOMs but less likely lead to orders of magnitude difference in yields.

### 3.3 High-NOx conditions

***Effects of NOx Level.*** The NO or NO₂ termination of $RO_2$ radicals competes with the $HO_2$ or $RO_2$ termination and forms nitrogen-containing species at the expense of other highly oxygenated products (Tsiligiannis et al., 2019; Garmash et al., 2020; Y. Wang et al., 2020; Mehra et al., 2020). Yet, quantitative understanding in the effects of NOx on the formation of oxygenated products is limited for aromatic precursors compared with those for biogenic VOCs (Nah et al., 2016; Lambe et al., 2017; Sarnela et al., 2018). The high-NOx experiments herein were conducted with $[NO_2]:[NO]$ ratios of 20 to 120 (Table S1), which

may represent urban afternoon conditions when fresh NOx emission is mostly converted to $NO_2$ and intense photochemistry fuels the oxidation of aromatics accompanying the NOx emission (Newland et al., 2021). Figure 4 shows the concentrations of observed HOMs for various $[NO_x]:[HO_2]$ conditions. We use $[NO_x]:[HO_2]$ instead of $[NO]:[HO_2]$ to evaluate termination pathways to form various nitrogen-containing products such as nitrated phenols, organic nitrate, and peroxynitrate. The contributions from $RO_2 + RO_2$ termination are probably minor because of the low concentrations of $RO_2$ (< 0.9 ppt). Given

the low and narrow range of NOx levels in the OFR254-iN₂O1.1 experiments, the product concentrations for all lamp voltages (i.e., a range of OH exposure) are averaged and shown as the first data point in each panel of Fig. 4.

    Similar to previous findings, nitrogen-containing products are the dominant species in the spectra with concentrations up to 18.3 and 7.3 ppt for benzene and toluene oxidation, respectively. The concentrations of all HOMs start to decrease at high $[NO_x]:[HO_2]$ ratios, except that the concentrations of dimeric products from benzene oxidation remain steady (Fig. 4a-b). The

decreasing trends for monomeric closed-shell, open-shell, and fragmented products are expected, indicating significant competition of radical terminations by NO or $NO_2$. The decreasing concentrations of nitrogen-containing products for increasing $[NO_x]:[HO_2]$ are however counterintuitive. Figure 4c-d shows the concentrations of main individual nitrogen-containing products. The initial increase of the concentrations of nitrogen-containing species might be explained by a decrease of $HO_2$ concentrations from 0.8 - 1.5 ppb to 0.5 - 0.7 ppb as a result of the switch of 1.1% of N₂O injection to 4.4% (Table S1).

The further reduction of these compounds as $[NO_x]:[HO_2]$ increases is probably related to the simultaneous decrease of $RO_2$, or alternatively further reactions to products that have two nitrogen atoms. Figure S8 in the Supplement shows similar concentration trends for individual ring-scission and ring-retaining products with or without nitrogen in their formulas. There seems to be "optimal" $[NO_x]:[HO_2]$ ratios of 130 to 240 for the formation of nitrogen-containing products. The dependence of those products with only one nitrogen atom on NOx is however not strong. The availability of $RO_2$ is perhaps the key factor

that limits their formation at low NOx levels and affects further reactions to form products with two nitrogen atoms at high NOx levels.

    Nitrated phenols are the most abundant nitrogen-containing products under high-NOx conditions. In aromatic oxidation, these compounds are formed by the reaction of phenoxy RO radicals with $NO_2$ (Jenkin et al., 2003). In the Master Chemical Mechanism (MCM v3.3.1), the phenoxy RO radicals can be formed via OH oxidation of phenols ($k_{OH} = 2.8 \times 10^{-11}$ cm³

molecule s⁻¹) with a low branching ratio of 0.06. They can also be formed by the NO₃ oxidation of phenols ($k_{NO_3} = 3.8 \times 10^{-12}$ cm³ molecule s⁻¹) with a high branching ratio of 0.74 (IUPAC, 2008). Under high-NOx conditions, the estimated concentrations

of OH and NO$_3$ radicals are 0.05 to 0.3 ppb and 0.01 to 0.09 ppb, respectively, suggesting that the NO$_3$ oxidation of phenols contributed efficiently to the formation of nitrated phenols in the OFR experiments herein. When [NO$_x$]:[HO$_2$] increases, the concentrations of C$_{6,7}$H$_{5,7}$NO$_4$ (perhaps nitrocatechol or methylnitrocatechol) increase first and then decrease. The concentrations of C$_{6,7}$H$_{5,7}$NO$_3$ (perhaps nitrophenol or methylnitrophenol) however show a weak dependence on the NO$_x$ level, suggesting the availability of RO radical might be the limiting factor in controlling the formation of nitrated phenols herein. Interestingly, the concentrations of products with two nitrogen atoms such as C$_6$H$_4$N$_2$O$_6$, C$_6$H$_8$N$_2$O$_9$, and C$_7$H$_{10}$N$_2$O$_9$ (only for toluene) steadily increase as [NO$_x$]:[HO$_2$] rises, suggesting a strong dependence of the formation of these species on NO$_2$.

***Formation of ROOH, RONO$_2$, and ROONO$_2$.*** Jenkin et al. (2019) suggested that the overall rate coefficients for RO$_2$ + HO$_2$ reactions are $1.92 \times 10^{-11}$ and $1.98 \times 10^{-11}$ cm$^3$ molecules$^{-1}$ s$^{-1}$ at 298 K for benzene and toluene oxidation, respectively. For the OFR conditions (Table S1), the characteristic time for the RO$_2$ termination by HO$_2$ was perhaps < 10 s, which is much shorter than the OFR residence time of 95 s. The rate coefficients of the hydroperoxide pathway (RO$_2$ + HO$_2$ → ROOH + O$_2$) may be constrained by the concentration ratios of [ROOH] multiplied by the loss rate of ROOH to [RO$_2$][HO$_2$] (Sect. S5 in the Supplement). For example, assuming that the C$_6$H$_7$O$_7$ and C$_7$H$_9$O$_7$ are the RO$_2$ radicals, and C$_6$H$_8$O$_7$ and C$_7$H$_{10}$O$_7$ are the corresponding ROOH for benzene and toluene oxidation, respectively, the slopes in Fig. 5a indicate that the rate coefficients of hydroperoxides are $1.20 \times 10^{-11}$ and $1.26 \times 10^{-11}$ cm$^3$ molecules$^{-1}$ s$^{-1}$. These rate coefficients suggest that the branching ratios of the hydroperoxide formation under low-NO$_x$ conditions are 0.62 and 0.64 for benzene- and toluene-derived RO$_2$, respectively, which are consistent with the estimated branching ratios of 0.52 - 1.00 in literature (Jenkin et al., 2019).

In the presence of NO$_x$, the reactions between C$_x$H$_{y+1}$O$_7$ and nitrogen oxides lead to both chain propagation and chain termination to form RO radicals and nitrogen-containing products. Similar to the analysis of the hydroperoxide formation, the slopes in Fig. 5b suggest that the rate coefficients of RO$_2$ + NO(+M) → RONO$_2$(+M) are $2.87 \times 10^{-11}$ and $6.12 \times 10^{-11}$ cm$^3$ molecules$^{-1}$ s$^{-1}$ for benzene and toluene oxidation under OFR254-5-iN$_2$O1.1 conditions, respectively (Sect. S5). These coefficients are more than one order of magnitude greater than the values reported by Jenkin et al. (2019) (i.e., $8.09 \times 10^{-13}$ and $1.10 - 8.45 \times 10^{-13}$ cm$^3$ molecules$^{-1}$ s$^{-1}$ for benzene and toluene oxidation, respectively). One explanation is that the detected C$_x$H$_{y+1}$NO$_8$ contains multifunctional groups and represents not only non-peroxy organic nitrates (RONO$_2$) but also peroxy organic nitrates (ROONO$_2$). As shown in Fig. 5c, the [C$_x$H$_{y+1}$NO$_8$]:[C$_x$H$_{y+2}$O$_7$] ratios start to decrease at higher [NO]:[HO$_2$] in our OFR254-5-iN$_2$O4.4 experiments. The competition between NO and HO$_2$ for terminating RO$_2$ should not alter the rate coefficients and the branching ratios (Atkinson and Arey, 2003). The lack of linear relationship of [C$_x$H$_{y+1}$NO$_8$]:[C$_x$H$_{y+2}$O$_7$] (i.e., assumed as [RONO$_2$]:[ROOH]) on [NO]:[HO$_2$] ratio is consistent with the possibility of C$_x$H$_{y+1}$NO$_8$ partially being ROONO$_2$, especially for toluene oxidation.

Xu et al. (2020) indicate that the formation of non-peroxy organic nitrates (RONO$_2$) is minor in aromatic oxidation. The detected ROONO$_2$ are likely RC(O)OONO$_2$, because that ROONO$_2$ are usually thermally unstable intermediates (Kirchner et al., 1999) and RC(O)OONO$_2$ can be detected by the NO$_3^-$-TOF-CIMS (Rissanen, 2018). Acylperoxy radicals RC(O)OO (i.e., a type of peroxy radical) may react with NO$_2$ to produce peroxyacylnitrate (RC(O)OONO$_2$) (Jenkin et al., 2019). The formation

of the $RC(O)OONO_2$ requires (1) the original $RO_2$ radicals to be $C_xH_{y+1}O_6$ instead of $C_xH_{y+1}O_7$ and (2) an acyl (-C=O) group that is connected to the O-O bond. The formation of $C_xH_{y+1}O_6$ is feasible through the RO pathway (Sect. S3). The hydroperoxides corresponding to $C_xH_{y+1}O_6$ are $C_xH_{y+2}O_6$ (i.e., $C_6H_8O_6$ and $C_7H_{10}O_6$ for benzene and toluene, respectively). Figure 5d-e shows the increase of $[C_xH_{y+1}NO_8]:[C_xH_{y+2}O_6]$ (i.e., perhaps $[RC(O)OONO_2]:[ROOH]$) for increasing $[NO_2]:[HO_2]$ ratios. In particular, the relationship between $[C_xH_{y+1}NO_8]:[C_xH_{y+2}O_6]$ and $[NO_2]:[HO_2]$ is nearly linear at high $[NO_2]:[HO_2]$ ratios in the OFR254-5-iN2O4.4 experiments. There is a lack of kinetic data for the reactions of $RC(O)OO + NO_2$ (Rissanen, 2018), which prevents us from further investigation.

## 4 Atmospheric Implications

In this study, we investigated the formation of HOMs in the OFR by the oxidation of benzene and toluene in a wide range of OH exposure and $NO_x$ conditions. Recent findings emphasize the significance of the bicyclic channel in the oxidation of light aromatics (Wang et al., 2017; Molteni et al., 2018; Tsiligiannis et al., 2019; Zaytsev et al., 2019b; Garmash et al., 2020; Y. Wang et al., 2020). The presence of $NO_x$ enhances the formation of organonitrates and even dinitrate organic compounds. Extensive autoxidation and accretion reaction has been reported for substituted aromatics (Y. Wang et al., 2020). Our results show enhanced formation of more-oxygenated products for elevated OH exposures. The formation of dimeric products however seems unfavorable under conditions of high OH exposure and low $NO_x$ for substituted aromatics. The suppression of dimeric products may affect the contribution of aromatic HOMs to new particle formation in the downwind of urban atmosphere. The changes of product distribution and concentration highlight the possibility that multi-generation OH oxidation proceed preferably via H subtraction rather than OH addition, although one HOM formula may correspond to various functionalities. For aged air masses, this may reduce the H:C ratios of HOM products from aromatic oxidation. Under high-$NO_x$ conditions, we show that the formation of products containing one nitrogen atom perhaps depend more significantly on the organic radicals (RO or $RO_2$) but less so on $NO_2$, while formation of products containing two nitrogen atoms depends significantly on $NO_2$. Further investigation on the roles of high-$NO_2$ conditions that represent a wide range of anthropogenically influenced environments in the oxidation of aromatic VOCs are needed. Moreover, we found that non-peroxy organic nitrates might form via $RO_2 + NO$ under low-$NO_2$ conditions; and $RO_2 + NO_2$ may dominate to produce $ROONO_2$ or $RC(O)OONO_2$ under high-$NO_2$ conditions. The reaction of $RC(O)OO$ with $NO_2$ to produce peroxyacylnitrates should be of particular importance with high $[NO_2]:[NO]$ ratios of tens to hundreds. Both of $ROONO_2$ and $RC(O)OONO_2$ are reservoirs of $RO_2$ radicals. Under conditions of high $[NO_2]:[NO]$ ratios (e.g., late afternoon in urban or suburban environments), the "effective" lifetimes of $RO_2$ radicals might become longer because of the formation of $RC(O)OONO_2$. Subsequent slow release of $RO_2$ radicals with the help of $NO_2$ may extend the formation of HOMs from VOC oxidation in urban environments to early evening when OH starts to decline and $NO_3$ has not yet built up, facilitating the development of regional SOA pollution.

*Data availability*. Data presented in this manuscript are available upon request to the corresponding author.

*Author contributions*. QC and YJL designed the study. XC conducted the experiments and data analysis with the help of YZ, KL, and GH. QC, YJL, and XC wrote the manuscript.

*Competing interests*. The authors declare no competing financial interests.

*Acknowledgments*. This work is supported by the MOST National Key R&D Program of China (2017YFC0213000, Task 3), the National Natural Science Foundation of China (41875165, 41961134034, 51861135102), the 111 Project of Urban Air

Pollution and Health Effects (B20009), The Science and Technology Development Fund, Macau SAR (File no. 0019/2020/A1), University of Macau (File no. MYRG2018-00006-FST). The authors gratefully acknowledge Tong Zhu, Ying Liu, Manjula R. Canagaratna, Andrew Lambe, and Peng Zhe for instrument support and helpful discussion.

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

**Figure 1.** Mass spectra of HOM products measured by the $NO_3^-$-TOF-CIMS for Exp. #2, #11, #16, and #26 in Table S1. (a): benzene, OFR254-5; (b) benzene, OFR254-5-iN$_2$O4.4; (c): toluene, OFR254-5; (d) toluene, OFR254-5-iN$_2$O4.4. The reagent ion $NO_3^-$ is omitted from the molecular formulas, whereas the *m/z* values refer to the mass-to-charge ratios of the fitted ions with $NO_3^-$. The relative intensities of ions having $m/z \geq 300$ are multiplied by 10.

**Figure 2.** Concentrations of fragmented, monomeric closed-shell, open-shell and dimeric products formed by benzene and toluene oxidation under low-NO$_x$ conditions (OFR254-5) at various OH exposures. For benzene oxidation, both of *x* and *y* are 6. For toluene oxidation, *x* is 7 and *y* is 8. BPR: bicyclic peroxy radical; HP: hydroperoxide; -C=O: carbonyl; -OH: alcohol.

**Figure 3.** Concentrations and relative contributions of closed-shell monomeric products formed by benzene and toluene oxidation under low-NO$_x$ conditions (OFR254-5) at various OH exposures. For benzene oxidation, both of *x* and *y* are 6. For toluene oxidation, *x* is 7 and *y* is 8.

**Figure 4.** Concentrations of fragmented, monomeric closed-shell, open-shell, dimeric and nitrogen-containing products formed by benzene and toluene oxidation under high-NO$_x$ conditions (OFR254-5-iN$_2$O1.1/4.4) at various [NO$_x$]:[HO$_2$] levels. Data for OFR254-5-iN$_2$O1.1 experiments were averaged and shown as the first data point in each panel. For benzene oxidation, both of *x* and *y* are 6. For toluene oxidation, *x* is 7 and *y* is 8.

**Figure 5.** Kinetic analysis on the formation of nitrogen-containing HOMs. (a): [ROOH]×$k_{loss}$ vs. [RO$_2$][HO$_2$]; (b-c) [RONO$_2$]:[ROOH] vs. [NO]:[HO$_2$]; (d-e) [RC(O)OONO$_2$]:[ROOH] vs. [NO$_2$]:[HO$_2$]. For benzene oxidation, both of *x* and *y* are 6. For toluene oxidation, *x* is 7 and *y* is 8. $k_{loss}$ represents the loss rate of corresponding HOM products in unit of s$^{-1}$.

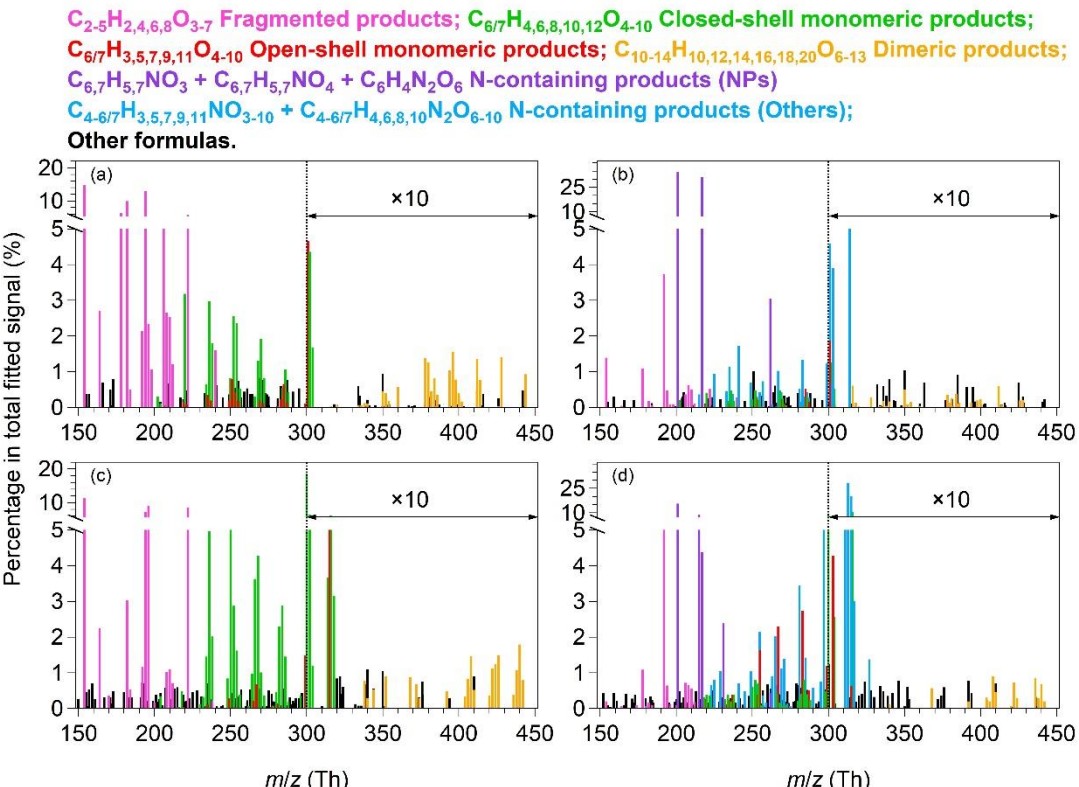

**Figure 1**

630

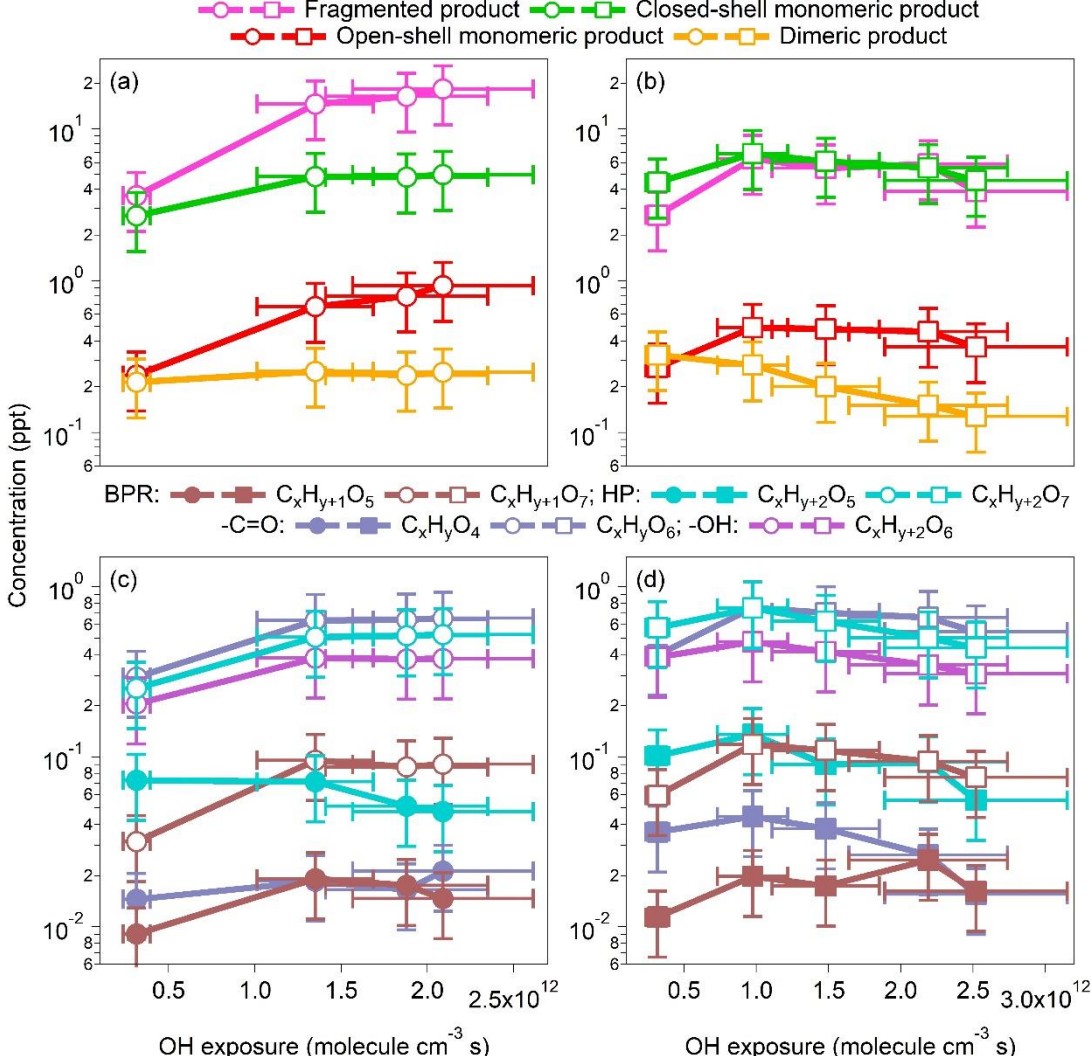

**Figure 2**

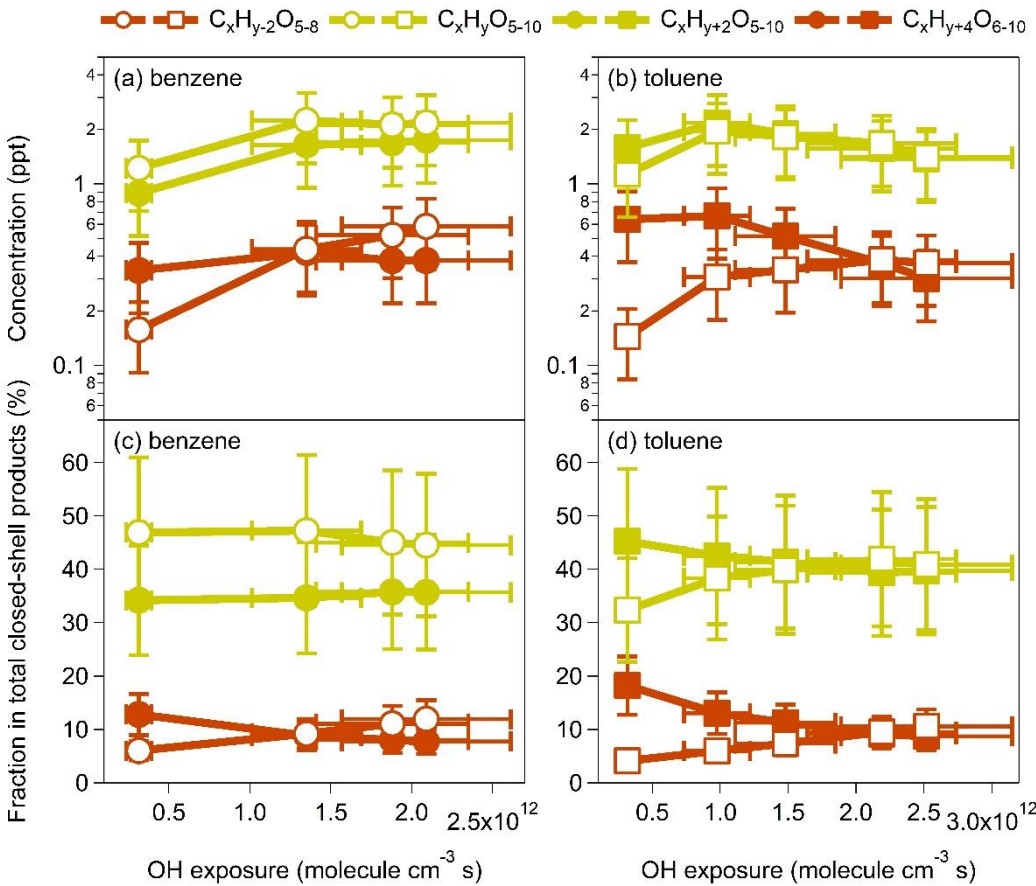

635 **Figure 3**

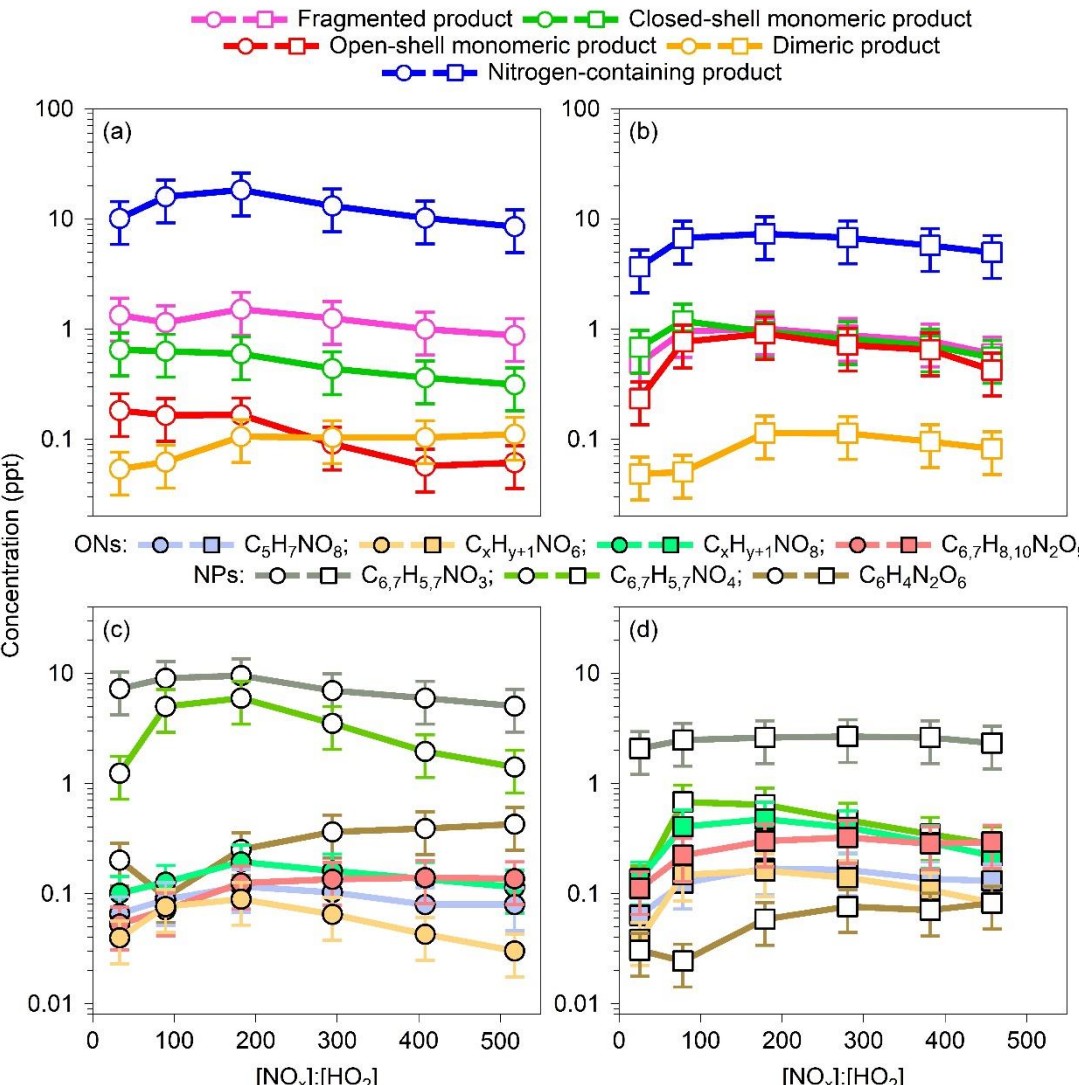

**Figure 4**

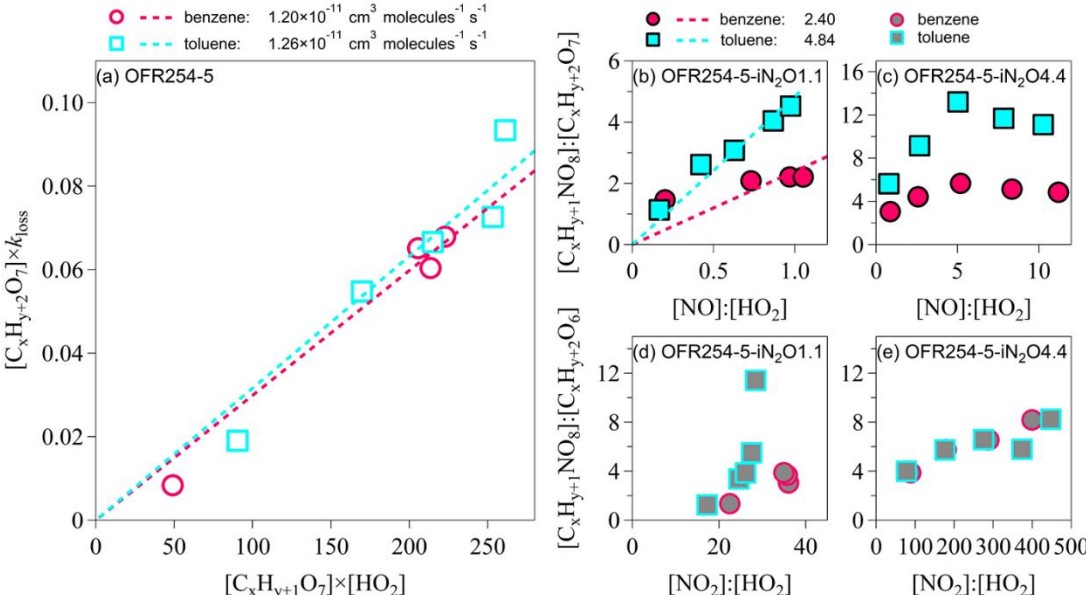

**Figure 5**