# Peer review of "S1. OH oxidation of benzene and toluene in the absence of NOx"

_Atmospheric Chemistry and Physics, 2021_

## Author Comment (AC1)

**Response to reviews**

We thank the reviewers for their constructive comments that help improve the manuscript. We provide below the point-by-point responses to those comments. Reviewer comments are in **bold**. Author responses are in plain text labeled with [R]. Line numbers in the responses correspond to those in the revised manuscript with all track changes accepted. Modifications to the manuscript are in *italics*.

**Reviewer #2**

**In this manuscript, Cheng et al., investigated the oxidation products of benzene and toluene using a PAM flow tube reactor under low- and high-$NO_x$ conditions. The authors used a nitrate-based tof-CIMS to measure the oxidation products (HOMs), and some unmeasured species, such as $HO_x$ and $RO_x$, were quantified with a chemical model designed for the PAM. By investigating the behavior of different HOM classes under different OH dose and $NO_x$ level, the authors suggested that 1) their system is more favorable for highly oxygenated products, 2) multi-generation OH oxidation is likely via the H-abstract route, and 3) Many even the majority of N-containing HOMs are likely peroxyacyl nitrates. Based on these results, some reasonable atmospheric implications were given. In general, I found this manuscript interesting and provided enough insights into the oxidation scheme for benzene and toluene. However, I do have some concerns, which should be addressed before it can be accepted for publication in ACP.**

[R0] We thank the reviewer for the valuable feedback. Detailed responses to the comments are given below.

**General comments:**
**How much the oxidation system of PAM with high OH concentration can be extrapolated to presented the real atmospheric conditions is always a major concern of the community. The lifetime of many effectively non-volatile HOMs is in the order of minutes, but the minimum equivalent OH exposure time in this study is 0.8 days.**

[R1] The purpose of OFR experiments is to explore the influence of various conditions on the product distributions and to some extent the formation mechanism of multi-generation oxidation of VOCs. While chamber experiments mimic better ambient environments, OFR experiments are easier to be set up for a wide range of conditions. Indeed, many of the HOMs observed in the OFR have been seen in ambient environment, although the OH exposure in the OFR is equivalent to several days of atmospheric photochemical age. For example, many of the detected isoprene nitrates during the SOAS 2013 campaign have been reported as the second- and third-generation OH oxidation products of isoprene under high-$NO_x$ conditions in the OFR experiments (Massoli et al., 2018). Also, the HOMs observed in the OFR from aromatic oxidation (including those

plausibly formed by multi-step OH oxidation) explain a significant portion of the HOMs observed in Beijing (unpublished results from the PKU and BUCT measurements).

**Also, at low OH concentrations, it is to be demonstrated if the ppm-level $O_3$ dominates the oxidation of the first-generation HOM products, which likely have an endocyclic double bond. Likewise, the $NO_3$-initated oxidation at high-$NO_x$ levels may also play an important role. These should be at least mentioned in the manuscript.**

[R2] The OH exposure in our experiments is about $1.1 \times 10^{11}$ to $2.5 \times 10^{12}$ molecules cm$^{-3}$. The OFR-based photochemical box simulations show that the aromatic oxidation reactions in our experimental conditions were dominated by OH rather than $O_3$ (Lambe et al., 2011; Peng et al., 2016). Similar to other OFR studies, we think the reaction rates of $O_3$ with oxidation products that contain double bounds are likely slower compared with that of OH (Molteni et al., 2018; Wang et al., 2020). The $NO_3$ concentrations in the OFR range from 0.01-0.09 ppb, for which the $NO_3$ oxidation of phenols may contribute efficiently to the formation of nitrated phenols in the OFR experiments because of the high branching ratio. Table R1 lists the general reaction rates of $RO_2$ with OH and $NO_3$ estimated for Exp. #28 (Jenkin et al., 2019). For such rates, the $NO_3$ reactions with HOM products might be minor under our experimental conditions compared with the OH reactions. To clarify the potential influence of $O_3$ and $NO_3$, we have added the above discussion in Line 89-96.

Table R1. The reaction of $RO_2$ with OH and $NO_3$ for Experiment #28

| Oxidants | Concentration (molecules cm$^{-3}$) | Rate coefficient (cm$^3$ molecules$^{-1}$ s$^{-1}$) | Rate (s$^{-1}$) |
|---|---|---|---|
| OH | $3.04 \times 10^9$ | $1.20 \times 10^{-10}$ | 0.4 |
| $NO_3$ | $2.21 \times 10^9$ | $2.40 \times 10^{-12}$ | 0.01 |

**The authors investigated the oxidation of benzene and toluene, and clear differences in the products and their response to different oxidation conditions were observed. However, readers might hope to see some more detailed explanation of these differences, rather than only some descriptions. However, it is ok if the authors show more dedicated studies in their follow-up manuscript.**

[R3] We thank the reviewer for the suggestion. The focus of this paper is the effects of oxidation conditions for a wide range of $HO_x$ and $NO_x$ levels on the product distributions. The formulae of the detected HOMs do not necessarily link to specific functionalities. It is therefore difficult to discuss much about the reasons of the difference between toluene and benzene results. We do have a follow-up manuscript in which we investigate the effects of alkyl substitution based on oxidation experiments of six different alkyl aromatics.

**Detailed comments:**
**L103-109. SP-AMS IS mentioned. However, no real data from this instrument was discussed in this manuscript.**

[R4] We used the Aerodyne LTOF-SP-AMS to measure the SOA mass concentrations for the calculation of the condensation sink. We have clarified this in Line 115 as follows: *"…(LTOF-SP-AMS) for the calculation of condensation sink in wall-loss corrections of HOMs"*.

**L125-127. Based on the observation, the authors suspected that another one or two steps of auto-oxidation may occur after BPRs form. This is an important observation, as it is different from what has been proposed by Wang et al., 2017, and thus may provide new insights. Can the authors propose a reaction scheme similar to Scheme S2?**

[R5] We have added Scheme S3 as an example for the proposed mechanism in the Supplement. Two potential routes for the further oxygen additions to the BPR follow the scheme proposed by Molteni et al. (2018) for mesitylene oxidation. One route represents the traditional autoxidation mechanism with internal H abstraction and oxygen addition as described by Wang et al. (2017). The other route involves cyclization forming a second oxygen bridge, which produces a carbon-centered radical followed with the addition of another oxygen molecule (Molteni et al., 2018). Toluene could undergo these two routes for the second step of auto-oxidation occurred after BPRs form because of the methyl group, which is different from benzene.

Scheme S3. The proposed mechanism of further autooxidation from the BPR $C_7H_9O_5$. Type I and Type II pathways are proposed by Wang et al. (2017) and Molteni et al. (2018).

**L165. It should be clarified that "Garmash et al. (2020) shows relatively high signals of $C_6H_8O_9$ and $C_{12}H_{14}O_8$ in the flow tube experiments, whereas…"**

[R6] Corrected.

**L171-172. The authors attribute the more steps of auto-oxidation in their experiments to the longer residence time. This is one possible reason. Can the authors exclude the possibility that the concentrations $HO_2$, $RO_2$, $NO_x$ in Garmash et al., (2020) were higher than those in this study, so that the auto-oxidation was suppressed to a greater extent?**

[R7] We agree with the reviewer about this possibility because the concentrations of $HO_2$ and $RO_2$ were not provided in the paper from Garmash et al. (2020). We have added a sentence in Line 202-204 as follows: *"Alternatively, the differences in $HO_2$ and $RO_2$ concentrations among different studies that remain unclear might affect the extent of auto-oxidation"*.

**L187-188. It is interesting to see that the dimeric HOMs decrease when OH exposure increases for toluene HOMs. It deserves a bit more explanation/speculation than just say "Whether this phenomenon is related to the substituted methyl group or not needs further investigations".**

[R8] As replied to the other reviewer's comment #12, we have added more discussion in Line 218-224 as follows: *"One potential contributor to the difference of the dependence of dimer formation on OH exposure is the more significant elevated $RO_2$ concentrations (0.2 to 0.9 pptv) as the OH exposure increases for benzene oxidation than those (0.3 to 0.5 pptv) for toluene oxidation while the $HO_2$ concentrations in the two sets of experiments are similar (1.5-2.4 ppbv). The enhancement of $RO_2$ concentrations may promote the dimer formation through the self or cross reactions of $RO_2$ (Mohr et al., 2017). On the other hand, a previous study indicates that the accretion of $RO_2$ depends on the functional groups of the $RO_2$ (Berndt et al., 2018). On the other hand, a previous study indicates that the accretion of $RO_2$ depends on the functional groups of the $RO_2$ (Berndt et al., 2018). Whether the decreasing concentrations of dimeric products with OH exposure is related to the steric effects of the substituted methyl group of toluene requires further investigations".*

**L255-260. The yield is probably one key message that readers would like to fetch from these studies. Thus, the big differences in the yields reported by different studies should be better explained. As the authors concluded that multi-generation oxidation is important in benzene and toluene oxidation, the highest yield by Garmash et al., (2020) and the lowest yield by Monteni et al., (2018) cannot be explained by the residence time, because the OH exposure in Monteni et al., (2018) is the highest among these studies. It may point to either the auto-oxidation is more important, which are different among these studies controlled by the termination reactions, or the peaks counted for "HOMs" are different. I can read from Table S4 that in Monteni et al., (2018), the OH dose and benzene concentration were highest, and thus the highest $RO_2$ concentration can be expected for that experiment, possibly leading to a termination of $RO_2$ auto-oxidation prematurely.**

[R9] The OH concentration listed in Table S4 for the study of Molteni et al. (2018) is the initial OH concentration, which should decrease significantly as the reaction proceeded. We have updated Table S4 for this information. Therefore, we cannot evaluate the actual OH exposure in the study of Molteni et al. (2018) and discuss further about the possible reasons. To clarify, we have added the following information in Line 304-308: *"The study of Molteni et al. (2018) only provided the initial OH concentration as listed in Table S4 that should decrease significantly as the reaction proceeds, for which we cannot rule out the possibility of low OH exposure that leads to fewer oxidation steps (i.e., lower yields of HOMs)."*

**L275. Why do the authors assume that $[NO_x]:[HO_2]$ should control the overall trend of**

**HOMs? It think k1[NO]+k2[NO$_2$]+k3[HO$_2$] is better than [NO$_x$]:[HO$_2$], because all of them lead to termination reactions that control HOM formation. Here, k1, k2, k3 is the average first-order rate constant of the bi-molecular reactions with RO$_2$.**

[R10] Here we focus on nitrogen-containing HOMs. [NO$_x$]:[HO$_2$] as well as [NO]:[HO$_2$] and [NO$_2$]:[HO$_2$] is chosen to differentiate the potential transition from RO$_2$+HO$_2$ to RO$_2$+NO/RO$_2$+NO$_2$ dominated regimes. The use of k$_1$[NO]+k$_2$[NO$_2$]+k$_3$[HO$_2$] is perhaps better for understanding the terminations but not our intention here.

**L294-295. Could NO$_3$ radical also be important in the formation of other HOMs, particularly for high-generation products?**

[R11] As replied in [R2], the NO$_3$ reactions with HOM products might be minor under our experimental conditions compared with the OH reactions. Autooxidation may occur at a rate of $10^{-3}$ to 1 s$^{-1}$ (Bianchi et al., 2019). Therefore, for RO$_2$ radicals that have high autooxidation rates, they may also preferably proceed with further autooxidation to form HOMs.

References:

Bianchi, F., Kurten, T., Riva, M., Mohr, C., Rissanen, M. P., Roldin, P., Berndt, T., Crounse, J. D., Wennberg, P. O., Mentel, T. F., Wildt, J., Junninen, H., Jokinen, T., Kulmala, M., Worsnop, D. R., Thornton, J. A., Donahue, N., Kjaergaard, H. G., and Ehn, M.: Highly oxygenated organic molecules (HOM) from gas-phase autoxidation involving peroxy radicals: a key contributor to atmospheric aerosol, Chem. Rev., 119, 3472-3509, https://doi.org/10.1021/acs.chemrev.8b00395, 2019.

Jenkin, M. E., Valorso, R., Aumont, B., and Rickard, A. R.: Estimation of rate coefficients and branching ratios for reactions of organic peroxy radicals for use in automated mechanism construction, Atmos. Chem. Phys., 19, 7691-7717, https://doi.org/10.5194/acp-19-7691-2019, 2019.

Lambe, A. T., Ahern, A. T., Williams, L. R., Slowik, J. G., Wong, J. P. S., Abbatt, J. P. D., Brune, W. H., Ng, N. L., Wright, J. P., Croasdale, D. R., Worsnop, D. R., Davidovits, P., and Onasch, T. B.: Characterization of aerosol photooxidation flow reactors: heterogeneous oxidation, secondary organic aerosol formation and cloud condensation nuclei activity measurements, Atmos. Meas. Tech., 4, 445-461, https://doi.org/10.5194/amt-4-445-2011, 2011.

Massoli, P., Stark, H., Canagaratna, M. R., Krechmer, J. E., Xu, L., Ng, N. L., Mauldin, R. L., Yan, C., Kimmel, J., Misztal, P. K., Jimenez, J. L., Jayne, J. T., and Worsnop, D. R.: Ambient measurements of highly oxidized gas-phase molecules during the Southern Oxidant and Aerosol Study (SOAS) 2013, ACS Earth Space Chem., 2, 653-672, https://doi.org/10.1021/acsearthspacechem.8b00028, 2018.

Molteni, U., Bianchi, F., Klein, F., El Haddad, I., Frege, C., Rossi, M. J., Dommen, J., and Baltensperger, U.: Formation of highly oxygenated organic molecules from aromatic compounds, Atmos. Chem. Phys., 18, 1909-1921, https://doi.org/10.5194/acp-18-1909-2018, 2018.

Peng, Z., Day, D. A., Ortega, A. M., Palm, B. B., Hu, W. W., Stark, H., Li, R., Tsigaridis, K., Brune, W. H., and Jimenez, J. L.: Non-OH chemistry in oxidation flow reactors for the study of atmospheric chemistry systematically examined by modeling, Atmos. Chem. Phys., 16, 4283-4305, https://doi.org/10.5194/acp-16-4283-2016, 2016.

Wang, S., Wu, R., Berndt, T., Ehn, M., and Wang, L.: Formation of highly oxidized radicals and multifunctional products from the atmospheric oxidation of alkylbenzenes, Environ. Sci. Technol., 51, 8442-8449, https://doi.org/10.1021/acs.est.7b02374, 2017.

Wang, Y., Mehra, A., Krechmer, J. E., Yang, G., Hu, X., Lu, Y., Lambe, A., Canagaratna, M., Chen, J., Worsnop, D., Coe, H., and Wang, L.: Oxygenated products formed from OH-initiated reactions of trimethylbenzene: autoxidation and accretion, Atmos. Chem. Phys., 20, 9563-9579, https://doi.org/10.5194/acp-20-9563-2020, 2020.

---

## Author Comment (AC2)

**Response to reviews**

We thank the reviewers for their constructive comments that help improve the manuscript. We provide below the point-by-point responses to those comments. Reviewer comments are in **bold**. Author responses are in plain text labeled with [R]. Line numbers in the responses correspond to those in the revised manuscript with all track changes accepted. Modifications to the manuscript are in *italics*.

**Reviewer #1**

**The manuscript by Cheng et al. investigates the formation of highly oxygenated organic molecules (HOMs) from the oxidation of benzene and toluene in a range of OH exposure and $NO_x$ conditions generated from flow-tube based experiments. Despite a recent discovery, HOMs have received increasing attention due to their high oxygenation and low volatility thereby constituting a widespread source of SOA in the atmosphere. A number of studies have suggested HOMs can be steadily produced through the autooxidation pathway from a number of hydrocarbon precursors of both biogenic and anthropogenic origins. This is a well-motivated and timely work that examines how environmental factors impact the HOMs product distributions from the photooxidation of benzene and toluene, two hydrocarbons that have received numerous investigations in the past, yet the crucial pathways leading to SOA formation remain uncertain despite decades of investigations. Some interesting observations are presented, such as the contrasting dimer formation kinetics between the benzene and toluene systems, and the optimal $[NO_x]:[HO_2]$ ratio that favors certain HOMs production, although the mechanism underlying these observations is elusive and warrants further evidence. Overall, it is a very detailed and interesting work. I recommend the manuscript for publication in Atmospheric Chemistry and Physics after the following comments being addressed in the revised version.**

[R0] We thank the reviewer for the valuable feedback. Detailed responses are given below.

**General comments:**
**My major comments are related to the interpretation of the trends and dynamics of the HOMs products in response to various flow tube conditions. Hundreds of HOMs products were identified in this work with the use of the 'inlet-less' nitrate CIMS that is highly sensitive to these 'sticky' molecules. It is however a great challenge to keep track of the trends and patterns of all these compounds without a good understanding of their chemical properties and formation mechanisms. I find the categorization of these compounds should be better clarified. The closed-shell products in general should include all stable products including both monomers and dimers. But it seems like here the authors use this term to specifically describe monomer products with a molecular formula of $C_xH_yO_z$. Perhaps 'closed-shell**

**monomer products' would better represent this category.**

[R1] We thank the reviewer for the suggestion. We have renamed the categories with "*closed-shell monomeric products*" and "*open-shell monomeric products*" based on the reviewer's suggestion.

**The open-shell products define all radical intermediates including both RO₂ and RO radicals. Whether these two types of radicals can be differentiated with the identified molecular formula needs to be clarified.**

[R2] In this study, the monomeric species having an odd number of hydrogen atoms could be organic peroxy radicals ($RO_2$) and alkoxy radicals (RO). However, as noted in Line 143-146, the open-shell monomeric products observed by the $NO_3^-$-TOF-CIMS in our experiments are more likely $RO_2$ radicals rather than RO radicals. This is because with relatively large carbon numbers, the former have lifetimes of seconds that are much longer than the latter of $< 10^{-4}$ s (Orlando et al., 2003; Seinfeld and Pandis, 2016; Zhao et al., 2018). We have revised Line 142-146 to make the point more clearly.

**Do these radicals ever possess higher carbon numbers (dimer-like type) than their hydrocarbon precursor?**

[R3] Yes, they may. For example, if H abstraction happens to a dimer that has an even number of hydrogen atoms, a dimer-like radical with an odd number of hydrogen atoms may form. A study in the boreal forest reported ~60 gaseous dimers ($C_{16-20}H_yO_{6-9}$) from monoterpene oxidation with odd or even numbers of hydrogen atoms (Mohr et al., 2017). In our study, the ion intensities of dimeric products with odd hydrogen atoms are very low. We therefore only focused on dimeric products with even hydrogen atoms such as $C_{10-14}H_{10,12,14,16,18,20}O_{6-13}$ (Figure 1) which are plausibly neutral molecules formed by hydrogen-bond dimerization or cross reactions of radicals (Zhao et al., 2018). We have clarified this part in Line 170-175.

**In general, the text should be clear in the formulations and definitions. While the readers appreciate categorization that certainly helps to reduce the complexity in interpreting the behaviors of hundreds of compounds, some unique information of individual species might get buried in the grouping process. I recommend the authors provide a list of the most abundant HOMs products that account for perhaps over half of the oxidized carbon in both benzene and toluene systems, along with the conditions that favor the production of each HOM compound. This would help at least elucidate dominant reaction mechanisms that govern the reaction fluxes.**

[R4] The peak list and relative signal contributions (%) of major gaseous HOMs produced by the benzene and toluene oxidation experiments under low and high $NO_x$ conditions are provided in the Supplement (Tables S2 and S3). We have revised the tables to specify the experimental conditions and revised Line 134 to clarify. The variations of selected products under different

conditions are discussed in detail in the main text related to Figures 2, 3, and 4.

**On a related note, have the authors thought about the potential of using PMF analysis to help to extract typical trends and patterns of different groups of HOMs compounds?**

[R5] Yes, positive matrix factorization (PMF) analysis was conducted on the unit-mass-resolution data for ions with mass-to-charge ratios ($m/z$) between 150 and 450 Th. As described in Sect. S2 in the Supplement, we used the PMF results to aid identification of product ions and non-product signals (Fig. S4). Three different product factors are identified. However, such temporal patterns provide limited information about the reaction kinetics and the formation mechanism of the products. So in this paper we mainly focus on some selected products that possibly represent specific reaction pathways to explore the oxidation mechanisms under a wide range of experimental conditions.

**Another concern I would like to share with the authors is whether the CIMS measurements constitute sufficient information for the assignment of different functionalities to a given molecular formula based on even/odd numbers of oxygen and hydrogen atoms. HOMs produced from the autooxidation pathway are generally recognized as multi-functionalized poly peroxides. The ring-retaining oxidation chemistry of aromatics further complicates the pathways that lead to the formation of oxygenated compounds. That being said, multiple combinations of different functional groups exist for any given molecular formula with high oxygen numbers. Have the authors conducted any collision induced dissociation experiments for the HOMs molecules and identified any characteristic fragments? Without further structural information of individual HOMs molecules, the categorization of certain molecular formulas as peroxides/alcohols here seems a little handwavy.**

[R6] The proposed product categories are based on (1) molecular formulas from reliable peak fitting, (2) traditional understanding of aromatic oxidation from literature, and (3) recent mechanistic developments (Mentel et al., 2015; Y. Wang et al., 2020; Garmash et al., 2020). For instance, in benzene/toluene ($C_xH_y$) oxidation by OH radicals, OH addition results in an alkyl radical with a formula $C_xH_{y+1}O_1$. This $C_xH_{y+1}O_1$ may quickly react with molecular oxygen ($O_2$) to a peroxy ($RO_2$) radical ($C_xH_{y+1}O_3$). The $RO_2$ radical $C_xH_{y+1}O_3$ can undergo endo-cyclization, where $RO_2$ attacks its own double bond to form an O-O bridge, resulting in a new (and more oxygenated) alkyl radical. This new alkyl radical reacts again with $O_2$ to form a bicyclic peroxy radical (BPR) $C_xH_{y+1}O_5$. In this pathway, compared with the parent VOC, addition of OH and two $O_2$ molecules changes the molecular composition by one H atom and five oxygen atoms. The BPR $C_xH_{y+1}O_5$ can undergo further autoxidation and form radicals with formula $C_xH_{y+1}O_7$. Autoxidation of $RO_2$ radicals involves intramolecular hydrogen shifts to the peroxide group from other carbon atoms and subsequent addition of oxygen to the produced carbon-centered radicals. The H shift itself does not modify the molecular composition, but the $O_2$ addition increases the oxygen content by an even number. Besides, BPR $C_xH_{y+1}O_5$ can form termination products of carbonyls ($C_xH_yO_4$),

alcohols ($C_xH_{y+2}O_4$), and hydroperoxides ($C_xH_{y+2}O_5$) (Mentel et al., 2015; Molteni et al., 2018). This means that one OH addition to the precursor $C_xH_y$ could lead to termination products with hydrogen numbers of $y$ and $y+2$. When the second OH attack occurs, products with hydrogen numbers of < y and > y+2 may be produced by multi-generation OH reactions (Table R1). With such established knowledge, some product formulae are associated with specific pathways. Some product formulae however may involve multiple pathways that we cannot differentiate (e.g., $y$, $y+1$, and $y+2$ products in Table R1). In this case, the collision induced dissociation (CID) experiments may be very helpful but we have not conducted such experiments to further explore possible structures of the HOM molecules like what have done in a recent study (Zaytsev et al., 2019a). Our discussion about the formation mechanisms are mainly made for those products that are associated with specific pathways. To clarify, we have included information regarding this in Line 120-132 and Line 265-267.

Table R1. Potential product formulae (oxygen number $\geqslant$ 5) from a second OH attack in benzene and toluene oxidation.

| Second OH attack | $H_y$- series products | | $H_{y+2}$- series products | |
| --- | --- | --- | --- | --- |
| | H abstraction | OH addition | H abstraction | OH addition |
| | Hydrogen number | | Hydrogen number | |
| Radical | $y-1$ | $y+1$ | $y+1$ | $y+3$ |
| Carbonyl | $y-2$ | $y$ | $y$ | $y+2$ |
| Alcohol | $y$ | $y+2$ | $y+2$ | $y+4$ |
| Hydroperoxide | $y$ | $y+2$ | $y+2$ | $y+4$ |

**I have a similar impression about whether the CIMS measurements alone are sufficient to drive the mechanisms discussed in the text, or rather using established mechanisms to rationalize the observations. For example, the authors highlight that 'multi-generation OH oxidation plays an important role in the product distribution, which likely proceeds more preferably via H subtraction than OH addition for early-generation products from light aromatics'.**

[R7] By applying the principles outlined in [R6] above, we may expect products with hydrogen number ($y$) shown in Table R1 from multi-generation OH reaction (e.g., the second OH attack) on the first-generation products from the initial reactions of VOCs with OH radicals. In multi-generation OH oxidation of the first-generation products, when the second OH attack occurs on a product that contains a hydrogen number of $y$ via H abstraction, termination products with hydrogen numbers $< y$ should be formed. Similarly, when the second OH attack occurs on a product that contains a hydrogen number of $y+2$ via OH addition, termination products with hydrogen numbers $> y+2$ should be formed. For OH addition to occur, the first-generation closed-shell molecules must still contain a carbon–carbon double bond in their structure (Garmash et al., 2020). The contributions of a third OH attack to the total of detected signals are often low (Molteni et al., 2018). We have added Table R1 into the Supplement as Table S5 and revised Line 265-267 and

Line 276-277 to clarify. We have also added in Line 289-290 to clarify that our interpretation is made on the basis of current mechanistic understanding of aromatic oxidation.

**Indeed, recent studies have proposed a unique pathway in the aromatic oxidation process, i.e., the ring-retaining chemistry that constantly takes up oxygen in the aromatic ring. The relevance and importance of this branch in the experimental conditions (e.g., high $HO_x$ intensities) of the present work needs to be explicitly discussed.**

[R8] Following the reviewer's suggestion, we have added discussions in the main text in Line 159-164: *"BPRs (i.e., $C_6H_7O_5$ from benzene oxidation and $C_7H_9O_5$ from toluene oxidation) can undergo unimolecular isomerization reactions to form more oxidized peroxy radicals, which compete with bimolecular reactions (Wang et al., 2017). The relative greater signal intensities of ring-retaining $O_7$ or $O_9$ HOM monomers compared with $O_{<6}$ ones suggest that the termination of $RO_2$ by $HO_2$ and the phenol oxidation pathway are perhaps relatively less important than the autooxidation pathway under experimental conditions herein (Calvert et al., 2002; Schwantes et al., 2017; Garmash et al., 2020)".* For the importance of phenol-pathway and auto-oxidation pathway, we add also added the following information in Sect.S1 in the Supplement as follows: *"Reactions of the newly formed phenols (e.g., cresol from toluene) can again be initiated by OH radicals, which have rate constants one order of magnitude higher than those of the aromatic precursors. Studies have shown that a major fraction of oxygenated compounds through this pathway has oxygen atoms less than 6 (Calvert et al., 2002; Schwantes et al., 2017; Garmash et al., 2020).*"

**In addition, the extent of OH exposure directly determines the number of $RO_2$ formed and consequently the likelihood of autooxidation that leads to HOMs production, but whether OH oxidation proceeds via abstraction or addition seems not directly supported by the observations presented in the text.**

[R9] As replied in [R7], products with hydrogen numbers of $< y$ are most likely formed via H abstraction, while those with hydrogen numbers of $> y+2$ are from OH addition on the basis of current mechanistic understanding (Table R1). We did not focus on the $y$, $y+1$, and $y+2$ products that may be involved in both H abstraction ad OH addition. Therefore, we think our results of the increasing fractions of $y-2$ products but decreasing fractions of $y+4$ products for increasing OH exposures as well as the monotonically increasing concentration of $y-2$ products may suggest that the multi-generation OH oxidation may proceed preferably via H subtraction than OH addition for early-generation products. Garmash et al. (2020) also noted that higher OH concentrations caused a larger variety in HOM products, with H-abstraction oxidation becoming more significant. We have added this point in Line 290-291 in addition to the revisions made in [R7].

**Specific comments:**
**- Line 15: This is a vague statement. Is there any unique feature of the OFR system used here**

**that likely promotes the HOMs formation compared with previous studies? or just simply based on the longer residence time / slower flow rate?**

[R10] The comparison of previous flow-tube experiments with our study are discussed in detail in Sect. 3.2, which includes several aspects not just simply based on the residence time. We have revised this sentence in abstract in Line 15 as follows: "*More oxygenated products present in our study than in previous flow-tube studies, highlighting the impact of experimental conditions on product distributions.*"

**- Line 185: It is interesting to see that 'the concentrations of fragmented, closed-shell and open-shell products first increase and then slightly decrease with the increasing OH exposure' for the toluene case. Have the authors thought about any potential fragmentation process that breaks these HOMs molecules apart?**

[R11] The fragmented products with carbon numbers less than their precursors may be formed by ring-scission (e.g., Scheme S2 in SI) that breaks the aromatic ring apart to form dicarbonyls and epoxy-dicarbonyl (Zaytsev et al., 2019b). In addition, some fragmented products can be formed through CO elimination from an acyl radical (Rissanen et al., 2014), splitting $CO_2$ from an RO radical (Garmash et al., 2020), or dealkylation (Birdsall and Elrod, 2011). We have added this information in Line 136-138.

**- Line 190: Responses of the dimer products to increasing OH oxidation of toluene was found different compared with the benzene system. While the presence of the methyl branch on the aromatic ring might play a role in fragmentation, it is ultimately the $RO_2$ self/cross combination process that results in the dimer formation. Have the authors compared the relative increases in both $RO_2$ and $HO_2$ levels in response to the increasing OH exposure in the benzene and toluene oxidation experiments?**

[R12] One potential contributor to the difference of the dependence of dimer formation on OH exposure is the more significant elevated $RO_2$ concentrations (0.2 to 0.9 pptv) as the OH exposure increases for benzene oxidation than those (0.3 to 0.5 pptv) for toluene oxidation while the $HO_2$ concentrations in the two sets of experiments are similar (1.5-2.4 ppbv). The enhancement of $RO_2$ concentrations may promote the dimer formation through the self or cross reactions of $RO_2$ (Mohr et al., 2017). On the other hand, a previous study indicates that the accretion of $RO_2$ depends on the functional groups of the $RO_2$ (Berndt et al., 2018). Whether the decreasing concentrations of dimeric products with OH exposure is related to the steric effects of the substituted methyl group of toluene requires further investigations. We have added the above discussion in Line 216-224.

**- Line 195-200: Functionalities cannot be simply inferred here solely based on the molecular formula without any additional structural information. Another layer of uncertainty that adds to the level of complexity is the rates and number of generations of autooxidation pathways.**

[R13] We agree with the reviewer that the interpretation of the closed-shell products are only possibilities. Given the added mechanistic information in the revised manuscript as replied in [R6], we know BPRs may form $C_xH_{y+2}O_{5,7}$ hydroperoxides, $C_xH_yO_{4,6}$ carbonyls, and $C_xH_{y+2}O_{4,6}$ alcohols. But the detected $C_xH_{y+2}O_{5,7}$ do not have to be hydroperoxides. We have revised the description here in Line 232-239 as follows: "*As described in Sect. 3.1, the further reactions of $C_xH_{y+1}O_5$ and $C_xH_{y+1}O_7$ may form closed-shell monomers such as hydroperoxides ($C_xH_{y+2}O_{5,7}$), carbonyls ($C_xH_yO_{4,6}$), and alcohols ($C_xH_{y+2}O_{4,6}$) (Mentel et al., 2015; Molteni et al., 2019), although these formulae may correspond to other functionalities depending on the reaction rates and numbers of generation of autooxidation pathways. For multiple steps of auto-oxidation of BPRs or $RO_2$ radicals ($C_xH_{y+1}O_5$ or $C_xH_{y+1}O_7$), the products with two more hydrogen atoms than the precursor are plausibly hydroperoxides ($C_xH_{y+2}O_z$) if z is an odd number and alcohols if an even number. On the other hand, products that have the same hydrogen atoms ($C_xH_yO_z$) are likely carbonyls with an even number of z. Instead, if the carbonyl formation involves the RO pathway, $C_xH_yO_z$ with an odd number of z may be formed.*"

**- Line 245: Another factor playing in here might be the much slower reaction rate of benzene toward OH radicals.**

[R14] We thank the reviewer for this suggestion. We have revised in Line 282-286 as follows: *"The decreasing trends at high OH exposures are more significant for toluene-derived products. Such differences might be explained by (1) the faster OH+VOC rate for toluene than for benzene and (2) the methyl group in toluene as well as the added -OH groups during oxidation that may increase the electron density on the aromatic ring through a resonant electron-donating effect and thereby activate the aromatic ring and facilitate further addition reactions to form products with multiple -OH groups (M. Wang et al., 2020)."*

**- Line 255: The authors need to provide more details about how the wall losses in flow tube and sampling lines are corrected in the molar yield calculations.**

[R15] The details for the wall loss corrections of HOMs are provided in Sect. S4 in the Supplement. We have added some information to Line 296-299 as follows: *"To calculate the molar yields of HOM products, wall losses are corrected, including the loss in the sampling line, the loss to the OFR walls estimated by eddy diffusion, the loss to aerosol particles in the OFR (i.e., the condensation sink calculated on the basis of particle measurements), and the loss to non-condensable products due to continuous reaction with OH (Palm et al., 2016). Details are provided in Sect. S4 in the Supplement."*

**- Line 282: How does the OH level change along with varying [NO$_x$]:[HO$_2$] ratios?**

[R16] The OH concentrations decreased from 0.19 ppb (1.1% vol $N_2O$) to 0.05-0.12 ppb (4.4% vol $N_2O$) for benzene and from 0.21 ppb (1.1% vol $N_2O$) to 0.05-0.12 ppb (4.4% vol $N_2O$) for toluene. This change of a factor of 2 to 4 is much smaller than the variation of [NO$_x$]:[HO$_2$] ratio

(i.e., a factor of 3 to 18).

**- Line 335: There have been a few numbers of recent studies demonstrating the formation of HOMs from the oxidation of alkyl-aromatics. This is a good place to summarize how the present study contribute to new insights into this system and what makes this study unique compared with previous studies.**

[R17] We thank the reviewer for the suggestion. We have added the following summary in Line 383-387: "*
[revised manuscript text omitted]